# Outcomes of Fenestrating vs. Reconstituting Laparoscopic Subtotal Cholecystectomy: A Single-Center Retrospective Study

**DOI:** 10.3390/healthcare13192465

**Published:** 2025-09-28

**Authors:** Abdullah Aloraini, Tariq Alanezi, Ahmad Madkhali

**Affiliations:** 1Division of General Surgery, Department of Surgery, College of Medicine, King Saud University, Riyadh 11451, Saudi Arabia; abdaloraini@ksu.edu.sa (A.A.); ahmadmadkhali@gmail.com (A.M.); 2Department of Surgery, University of Toronto, Toronto, ON M5S 1A1, Canada

**Keywords:** cholecystectomy, subtotal, laparoscopic, fenestrating, reconstituting, bailout cholecystectomy, gallbladder, surgery

## Abstract

**Background**: Laparoscopic subtotal cholecystectomy (LSTC), either fenestrating or reconstituting, offers potential benefits for the “difficult gallbladders” in terms of reduced bile duct injury (BDI) risk. **Methods**: This single-center retrospective cohort study analyzed data from patients who underwent reconstituting or fenestrating LSTC at a tertiary care center. We excluded patients who were converted to open cholecystectomy or had incomplete medical records. The data examined included demographic and clinical characteristics, preoperative assessments, operative details, and postoperative outcomes. No multivariable regression was performed because of the limited sample size. **Results**: The study included 46 patients (reconstituting: 20 patients; fenestrating: 26 patients). The severity of cholecystitis assessed by the Tokyo guidelines showed a higher proportion of Grade 2 severity in the reconstituting group than the fenestrating group (90% vs. 56.5%; *p* = 0.027). Both surgical techniques were similarly challenging and showed no significant differences in operative difficulty, operative duration, blood loss, or total hospital stay. Fenestrating procedures had non-significantly higher incidences of BDI (7.7% vs. 0%; *p* = 0.21), bile leakage (23.1% vs. 10%; *p* = 0.246), and intraoperative drain placement (88.5% vs. 75%; *p* = 0.232). Postoperative complications such as bile leaks were also comparable between the two techniques. Nevertheless, given the small sample, these observations are descriptive and should not be interpreted as evidence of comparability or superiority. **Conclusions**: Despite limitations, our analysis suggests that fenestrating and reconstituting approaches have comparable postoperative outcomes, although fenestrating procedures were associated with slight but non-significant increases in BDI and drain placement due to leaks. The choice of LSTC technique should depend on intraoperative findings, surgical expertise, and familiarity with each technique, but further studies are needed to obtain firm conclusions.

## 1. Introduction

Laparoscopic cholecystectomy (LC) is currently the standard approach for gallbladder diseases but can involve significant risks. These risks are especially notable in challenging cases where the anatomy is distorted or where the gallbladder is severely inflamed, fibrotic, or adherent due to chronic cholecystitis or previous episodes of acute inflammation [1,2]. Bile duct injuries (BDIs) are potentially serious complications of LC in patients with difficult gallbladders [3]. Although these injuries are relatively uncommon, they can lead to severe consequences, such as biliary leakage, cholangitis, biliary cirrhosis, and the need for complex reconstructive surgeries, which significantly impair quality of life and prolong hospital stays [4,5,6]. Moreover, BDIs are associated with a higher risk of postoperative mortality and increased utilization of healthcare resources [7].

Conventionally, open cholecystectomy has been regarded as the gold standard for managing difficult gallbladders, and the primary reason is that it allows for a better critical view, which improves safety [8]. Despite its advantages in terms of direct visualization, open cholecystectomy involves larger incisions, resulting in increased postoperative pain, longer recovery periods, and greater overall morbidity than laparoscopic approaches [9]. In recent years, there has been a shift towards laparoscopic techniques, even for complex cases, and laparoscopic subtotal cholecystectomy (LSTC) has been gaining prominence as a viable alternative when open cholecystectomy is risky or unfeasible. LSTC offers several advantages, including the benefits of a minimally invasive approach while potentially reducing the risk of BDI [10,11]. Studies have demonstrated that LSTC leads to lower risks of BDI and shorter hospital stay than open cholecystectomy [12]. However, it can increase the risk of bile leaks from the remnant gallbladder [13,14].

Currently, two techniques are commonly described for LSTC: fenestrating and reconstituting LSTC. Fenestrating LSTC avoids the creation of a closed remnant but may increase the risk of postoperative bile leakage from the open gallbladder base. In contrast, reconstituting LSTC seals the gallbladder remnant and may reduce immediate bile leakage, but it leaves a pathological remnant that can act as a nidus for recurrent stone formation and recurrent biliary symptoms over time. Specifically, the reconstitution often results in a remnant that could meet Bodvall and Overgaard’s criteria for a gallbladder remnant if an endoscopic retrograde cholangiopancreatography (ERCP) were to be performed immediately post-operation [15,16].

Reconstituting the gallbladder at Hartmann’s pouch involves creating a remnant to reduce the incidence of postoperative bile fistulas, which are more common with the fenestrating technique [1,17]. Specific procedural differences include the management of the cystic duct and gallbladder lumen closure. The fenestrating technique leaves the posterior wall of the gallbladder or a portion of the gallbladder adherent to the liver in situ, while the reconstituting technique employs sutures or staplers to close the lumen [16]. Therefore, the long-term risk of symptomatic remnant disease is a key concern with the reconstituting approach.

The choice between fenestrating and reconstituting techniques is not straightforward. Some of the reasons for this are the lack of prospective head-to-head comparative studies and the insufficient data on the long-term incidence of symptomatic gallbladder remnants following these surgeries. Therefore, this single-center retrospective cohort study assessed intra- and postoperative outcomes of reconstituting and fenestrating LSTC at a single tertiary care center in Saudi Arabia.

## 2. Materials and Methods

### 2.1. Study Design and Data Source

The study protocol was approved by the Institutional Review Board of the College of Medicine, King Saud University (Reference No. E-24-8663). Given the retrospective nature of the study, the requirement for informed consent was waived. This retrospective cohort study was conducted at a tertiary care center to assess the outcomes of reconstituting versus fenestrating LSTC. The study population consisted of adult patients (aged > 18 years) who underwent LSTC at King Khalid University Hospital between January 2016 and June 2023.

Patients were deemed eligible if they had an intraoperative diagnosis of complicated gallbladder disease, including acute cholecystitis, chronic cholecystitis with severe fibrosis, and gallbladders that are difficult to remove due to anatomical distortions or extensive adhesions [18]. The included patients were limited to those who underwent either reconstituting or fenestrating LSTC. The decision for the type of subtotal cholecystectomy performed was based on intraoperative findings and surgeon’s discretion/expertise. Patients were excluded if they had incomplete medical records or malignant gallbladder disease. Moreover, we excluded cases that were converted to open cholecystectomy as the objective of the analysis was to describe outcomes of LSTC. We acknowledge that this exclusion may have reduced the representation of the most severe cases and therefore introduces selection bias.

The two LSTC subtypes were described as “fenestrating” (open remnant without restoration of lumen) and “reconstituting” (closure of the remnant, which is also referred to here as “reconstituting [closure] LSTC” or “closure after partial resection”). Most procedures were performed by a consistent team of hepatobiliary surgeons, although some were performed by other general surgeons who had different levels of expertise but had substantial experience in laparoscopic cholecystectomies and laparoscopic procedures. For patients undergoing the fenestrating technique, the cystic duct was internally sutured in all cases to minimize the risk of postoperative bile leakage.

### 2.2. Data Collection

Data were extracted for a wide range of clinical variables, including preoperative, intraoperative, and postoperative data. Preoperative data involved demographic details such as age, sex, and body mass index (BMI). Clinical assessments included the American Society of Anesthesiologists (ASA) physical status classification, smoking history, and existing comorbid conditions. Patients’ surgical history was reviewed for any previous bariatric procedures and previous episodes of biliary disorders like cholecystitis or cholangitis. The severity of cholecystitis was evaluated based on the Tokyo Guidelines 2018 (TG18) [19], and surgical urgency was noted. The complexity of the anticipated surgery was quantified using the Randhawa Scoring Group score [20] and Tongyoo Scoring Group score [21], which classify operative difficulty on a scale ranging from easy (scores of 1–5) to very difficult (scores of 11–15).

Additionally, the gallbladder wall thickness was determined based on preoperative radiological imaging (ultrasound or computed tomography (CT) and categorized using four groups ranging from normal (1–2 mm) to severely thickened (7 mm or more) [22]. The operative difficulty was graded as easy, difficult, or very difficult according to the criteria presented in Table 1. Operative difficulty scores were recorded prospectively in the operative notes for all attempted laparoscopic cholecystectomies, and the analysis included only patients whose LSTC was completed laparoscopically. Consequently, the proportions of “very difficult” cases reported here exclude converted cases and may not reflect the full spectrum of difficulty encountered during attempted laparoscopic cholecystectomy.

The examined intraoperative data included the duration of the surgery, estimated blood loss, occurrence of BDIs, and use of intraoperative drains. Postoperative data included the occurrence of complications such as bile leaks, collections of fluid around the gallbladder, and any mortality. The postoperative metrics also included the duration of hospital stays and rates of readmission or further interventions within the first 30 days following the surgery.

### 2.3. Statistical Analysis

The statistical analysis was conducted using R software for Windows (R version 4.3.1). Continuous variables such as age, BMI, operative duration, and blood loss were expressed as the mean ± standard deviation (SD) for normally distributed data and the median with interquartile range for data that lacked a normal distribution. Categorical variables, including sex, ASA classification, and postoperative complications, were reported as frequencies and percentages.

Comparative analysis between the two groups (reconstituting versus fenestrating LSTC) was performed using the chi-squared test or Fisher’s exact test for categorical variables. For continuous variables, an independent-samples t-test was employed for normally distributed data, while the Mann–Whitney U test was used for data that did not follow a normal distribution. The level of significance was set at a *p*-value of less than 0.05 for all statistical tests. No multivariable regression was performed because of the limited sample size, which means that we could not exclude residual confounding.

## 3. Results

A total of 46 patients underwent LSCT during the data-collection period (reconstituting: 20 patients; fenestrating: 26 patients). The mean age of patients was relatively comparable between the fenestrating group (50.3 ± 17.9 years) and the reconstituting group (46.4 ± 15.4 years), and the difference was not significant (*p* = 0.393). Gender distribution also showed no significant difference (*p* = 0.474). Similarly, BMI values were not significantly different with means of 30.8 ± 6.4 and 28.7 ± 6.5 kg/m^2^ for the fenestrating and reconstituting groups (*p* = 0.412), respectively.

Significant differences were observed for smoking status: 15% of the reconstituting group were current smokers, while there were no current smokers in the fenestrating group (*p* = 0.043). No significant differences were detected in the prevalence of comorbidities between the two groups (*p* > 0.05). A significant difference was observed in the Tokyo Guideline grading: a significant proportion of the reconstituting group were Grade 2 or 3 (90%) compared to only 65.2% in the fenestrating group (*p* = 0.027; Table 2).

None of the included patients underwent preoperative endoscopic retrograde cholangiopancreatography or percutaneous transhepatic cholangiography for biliary obstruction. There was no statistically significant difference in the occurrence of a radiologically contracted gallbladder between the fenestrating group (40%) and the reconstituting group (31.6%; *p* = 0.796). However, the incidence of irregular gallbladder walls was significantly higher in the fenestrating group (7.7%) than the reconstituting group (0%; *p* = 0.045). The presence of a clinically palpable gallbladder was rare and was identified in only one patient in the reconstituting group (5%). The difference between groups in this regard was not statistically significant (*p* = 0.246).

The gallbladder wall thickness showed no significant difference between the two groups (*p* = 0.292). Similarly, comparisons of laboratory values revealed no statistically significant differences. The mean international normalized ratio (INR) also showed no significant difference between groups (*p* = 0.586; Table 3).

Regarding preoperative assessment of preoperative difficulty, the results showed no statistically significant association between the Randhawa scoring groups and the type of surgical procedure (*p* = 0.762). Nearly 38.5% of the fenestrating group and 25% of the reconstituting group had a Randhawa score of 6–10. In the case of Tongyoo scoring groups, there was also no statistically significant association with the type of surgical procedure (*p* = 0.393). Furthermore, 57.7% of the fenestrating group and 65% of the reconstituting group had a Tongyoo score of 6–10 (Figure 1).

Regarding operative difficulty, there was no statistically significant difference between the fenestrating and reconstituting groups (*p* = 0.784). Specifically, 23.1% of fenestrating procedures were categorized as difficult compared to 20% in the reconstituting group. Larger proportions in both groups were considered very difficult (66.9% for the fenestrating group and 80% for the reconstituting group). The average hospital stay was similar between the groups (9 days in the fenestrating group and 5 days in the reconstituting group), but the difference did not reach statistical significance (*p* = 0.977).

The operation duration and blood loss were similar between the two groups. The mean operative duration was 181.7 min for the fenestrating group and 191.8 min for the reconstituting group (*p* = 0.650), while the mean blood loss was 153.9 mL for the fenestrating group and 121.7 mL for the reconstituting group (*p* = 0.347). There was a notable but statistically insignificant difference in the incidence of BDI (beyond bismuth type 1 BDI), with the fenestrating group having a 7.7% incidence, while the reconstituting group had a 0% incidence (*p* = 0.21). The rate of intraoperative drain placement was slightly higher in the fenestrating group (88.5%) than the reconstituting group (75%), although this difference did not reach statistical significance (*p* = 0.232).

The incidence of peri-cholecystic fluid collection was higher in the reconstituting group (40%) than in the fenestrating group (23.1%), but these differences were not statistically significant (*p* = 0.216). The incidence of bile leaks was numerically higher in the fenestrating group (23.1%) than in the reconstituting group (10%), but again with no statistical significance (*p* = 0.246). The incidence of a previous history of biliary inflammation (including cholecystitis and cholangitis) was similar between the groups with 26.9% in the fenestrating group and 40% in the reconstituting group (*p* = 0.348; Table 4). Three patients in the reconstituting group had recurrence of symptoms postoperatively, while none occurred in the fenestrating group; however, no surgical reintervention was performed (*p* > 0.05).

## 4. Discussion

LSTC has emerged as an important strategy for managing challenging gallbladder cases, particularly when anatomical distortions or severe inflammation complicate a standard cholecystectomy [13,25]. This procedure can be executed using either the “fenestrating” or “reconstituting” approaches, which both have unique benefits and challenges. The fenestrating technique, which leaves the gallbladder base open and may include suturing of the cystic duct internally, can potentially lead to increased risks of postoperative biliary fistulas due to bile leak [26]. In contrast, the reconstituting method involves sealing off the lower end of the gallbladder, which mitigates the risk of postoperative fistulas, but this method results in the formation of a gallbladder remnant, which may potentially cause recurrent symptoms [27]. In the present study, we analyzed the outcomes of fenestrating versus reconstituting LSTC at a single tertiary care center.

The results showed that the incidence of bile leakage was numerically higher in the fenestrating group (23.1%) than in the reconstituting group (10%). Although this difference did not reach statistical significance (*p* = 0.246), it suggests a trend that warrants attention. As mentioned, the fenestrating approach inherently leaves a part of the gallbladder wall open to the peritoneal cavity, which could potentially facilitate bile leakage. In a recent meta-analysis, fenestrating LSTC was associated with a higher risk of bile leak than reconstituting LSTC [26]. Similarly, a retrospective chart review of 191 patients who underwent subtotal cholecystectomy showed that the fenestrating technique was associated with a higher risk of this complication [13]. Other studies have shown similar findings [27,28].

In the present analysis, the absence of statistical significance between the groups regarding bile leakage incidence might have been influenced by the relatively small sample size and the variability in surgical technique and expertise. Additionally, the management of the cystic duct during these procedures can vary significantly and may also impact the outcomes related to bile leakage. Thus, further research with larger sample sizes and standardized surgical protocols could help to clarify the relationship between surgical technique and the risk of bile leakage.

Three patients in the reconstituting group developed recurrence of their symptoms, but no surgical reintervention was performed. These findings are in line with meta-analyses and retrospective studies showing comparable rates of recurring symptoms between fenestrating and reconstituting LSTC [13,26,27,28]. The absence of symptoms in the short term is encouraging, yet the potential for long-term issues, particularly with gallbladder remnants in the reconstituting group, remains a concern that is not fully addressed by immediate postoperative outcomes. Future research should focus on longitudinal studies with longer follow-up durations.

BDI remains one of the most serious complications of gallbladder surgery and can potentially have grave consequences for patient outcomes and healthcare resources [29]. In a recent meta-analysis comparing the outcomes of LSTC versus open cholecystectomy for difficult gallbladders, both fenestrating and reconstituting LSTC were associated with significantly lower rates of BDI, shorter operative duration, and reduced total hospital stay [30]. Similar findings were also found in other recent studies comparing the outcomes of open versus laparoscopic techniques [31]. The present study indicated a slightly higher rate of BDI in the fenestrating group (7.7%), although this difference was not statistically significant (*p* = 0.21). Similarly, operative duration, blood loss, and total hospital stay were comparable between the two groups.

### Study Limitations

Although the present study provides valuable insights, it has some limitations that must be acknowledged. First, the retrospective nature of the study inherently limits our ability to control for all potential confounding variables that could influence the outcomes. Reliance on pre-existing medical records may introduce information bias and result in missing or inconsistently recorded data. For example, the time interval between the onset of symptoms and the surgical procedure was not consistently documented in the medical records, so it could not be analyzed. This is a relevant limitation since delayed surgery may influence operative difficulty and postoperative outcomes. Additionally, this study was underpowered due to the limited sample size, which increased the risk of type II errors. Given the retrospective single-center design and surgeon-dependent decision-making, confounding by indication is possible.

Other relevant clinical details that could influence patient-centered outcomes were lacking, such as postoperative pain, return to normal function, and health-related quality of life. Moreover, while the surgical techniques were generally well defined, intraoperative variables were not standardized, including gallbladder remnant closure, cystic duct management, and drain usage. These variables likely differed between cases and were judged on a case-by-case basis, which could have introduced variability that could have influenced outcomes. Additionally, although the mean follow-up duration was 46.5 months, the SD was substantial (68.1 months), indicating a wide variation in follow-up length. This may have resulted in underreporting of late complications such as remnant inflammation, gallstone recurrence, or persistent symptoms.

Significantly, no imaging or endoscopic follow-up was reported in regard to assessment of the status of the gallbladder remnant in the reconstituting group. As such, long-term complications like stone formation, inflammation, or neoplastic transformation remain unexplored in this cohort and should be a focus of future longitudinal studies. The incomplete documentation in the medical records limits the assessment of the true impact of each surgical approach. Future prospective studies should aim to systematically capture and analyze these variables.

Additionally, the relatively small sample size might have affected the statistical power to detect significant differences between the groups and may have led to type II errors. For example, although the history of previous biliary inflammation was documented, the small sample size limited the performance of a stratified analysis. Given the small sample size, it was not feasible to perform a multivariable regression to adjust for confounders (e.g., BMI, ASA class, gallbladder wall thickness, severity grading), and residual confounding could not be ruled out.

Since the choice of the technique was based on intraoperative judgment and surgeon preference rather than on predefined criteria or randomization, a potential risk of confounding by indication was introduced. In other words, more complex cases may have been preferentially assigned to one technique over the other, which would have influenced outcomes independently of the surgical approach itself. Another limitation is the single-center design, which may limit the generalizability of the findings to other settings with different patient demographics or surgical expertise. Furthermore, practices and outcomes at a tertiary care center might not reflect those at other hospitals or in different healthcare systems. For these reasons, our results should be interpreted as descriptive or for the generation of hypotheses and not as definitive evidence of equivalence or superiority between these techniques.

## 5. Conclusions

This single-center retrospective study suggests that both reconstituting and fenestrating LSTC are viable bailout options for managing difficult gallbladders. The fenestrating approach showed a numerically higher incidence of BDI and bile leakage, and the reconstituting approach carried risks of symptom recurrence, but no significant differences were detected. The choice between techniques should be guided by intraoperative findings and surgical expertise. Nevertheless, because of the retrospective design, small sample size, and lack of randomization, no definitive conclusions regarding equivalence or superiority between the two techniques can be drawn. Therefore, larger multicenter prospective studies are needed before definitive comparisons can be made.

## Figures and Tables

**Figure 1 healthcare-13-02465-f001:**
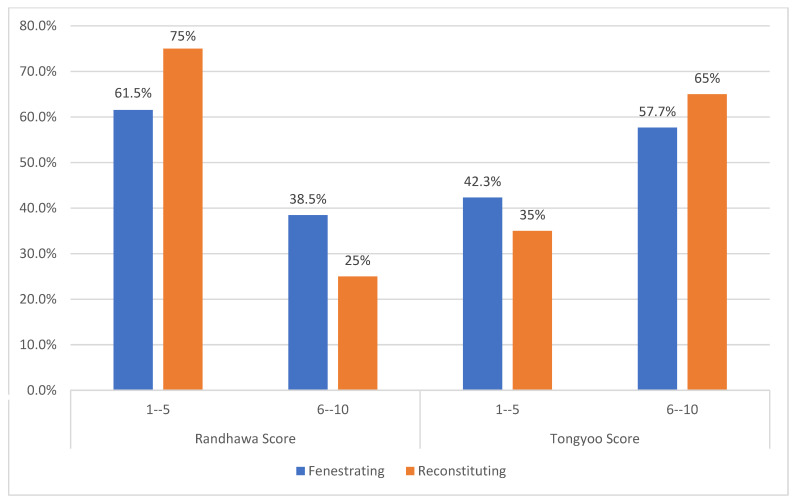
Preoperative difficulty scores.

**Table 1 healthcare-13-02465-t001:** Grading of operative difficulty [21,23,24].

Grade	Parameters
Easy	Time taken: <60 min
	No bile spillage
	No injury to the duct, artery
Difficult	Time taken: 60–120 min
	Bile/stone spillage
	Injury to the bile duct
Very difficult	Time taken: >120 min

**Table 2 healthcare-13-02465-t002:** Demographic and clinical characteristics of the included patients.

	Fenestrating (*n* = 26)	Reconstituting (*n* = 20)	Total (*n* = 46)	*p*-Value
Age, Mean (SD)	50.3 (17.9)	46.4 (15.4)	48.6 (16.8)	0.393
Sex				0.474
Female	9 (34.6%)	9 (45%)	18 (39.1%)	
Male	17 (65.4%)	11 (55%)	28 (60.9%)	
BMI, Mean (SD)	30.8 (6.4)	28.7 (6.5)	29.9 (6.5)	0.412
Smoker Status				0.043
Ex-smoker	1 (3.8%)	0	1 (2.2%)	
Non-Smoker	25 (96.2%)	17 (85%)	42 (91.3%)	
Smoker	0	3 (15%)	3 (6.5%)	
HTN	7 (26.9%)	7 (35%)	14 (30.4%)	0.75
DM	9 (34.6%)	7 (35%)	16 (34.8%)	0.99
DLP	1 (3.8%)	3 (15%)	4 (8.7%)	0.303
CAD	1 (3.8%)	0	1 (2.2%)	0.99
Acute Pancreatitis	1 (3.8%)	0	1 (2.2%)	0.99
Liver Disease	0	1 (5%)	1 (2.2%)	0.44
Renal Insufficiency	1 (3.8%)	1 (5%)	2 (4.4%)	0.99
Other	3 (11.5%)	2 (10%)	5 (10.8%)	0.99
Cholecystitis Type				0.37
Acute	9 (34.6%)	8 (40%)	17 (37%)	
Acute on Chronic	10 (38.5%)	4 (20%)	14 (30.4%)	
Chronic	7 (26.9%)	8 (40%)	15 (32.6%)	
Operative Priority				0.77
Elective	12 (46.2%)	11 (55%)	23 (50%)	
Emergency	14 (53.8%)	9 (45%)	23 (50%)	
ASA Group				0.621
1 or 2	22 (88%)	19 (95%)	41 (89.1%)	
3 or 4	4 (12%)	1 (5%)	5 (10.8%)	
Tokyo Guidelines Grading				0.027
Grade 1	9 (34.6%)	2 (10%)	11 (23.9%)	
Grade 2	15 (57.7%)	18 (90%)	33 (71.7%)	
Grade 3	2 (7.7%)	0	2 (4.3%)	

ASA: American Society of Anesthesiologists; BMI: Body mass index; CAD: Coronary artery disease; DLP: Dyslipidemia; DM: Diabetes mellitus; HTN: Hypertension.

**Table 3 healthcare-13-02465-t003:** Clinical, radiological, and laboratory characteristics of the included patients.

	Fenestrating (*n* = 26)	Reconstituting (*n* = 20)	Total (*n* = 46)	*p*-Value
Radiologically contracted gallbladder? *	10 (38.5%)	6 (30%)	16 (34.7%)	0.796
Irregular gallbladder wall? *	2 (7.7%)	0	2 (4.4%)	0.045
Clinically palpable gallbladder?	0	1 (5%)	1 (2.2%)	0.246
Gallbladder Wall Thickness (mm) Classification **				
N-Miss	0	1 (5%)	1 (2.2%)	0.292
Normal	2 (7.7%)	4 (20%)	6 (13%)	
Mild	13 (50%)	8 (40%)	21 (45.6%)	
Moderate	3 (11.5%)	3 (15%)	6 (13%)	
Severe	8 (30.8%)	4 (20%)	12 (26.1%)	
Laboratory ***				
Total Bilirubin, Mean (SD)	12.2 (9.7)	12.7 (9.4)	12.4 (9.4)	0.931
Direct Bilirubin, Mean (SD)	8.7 (18.9)	8.5 (14.1)	8.6 (16.9)	0.682
ALP, Mean (SD)	149.2 (150.8)	187.3 (161.5)	165.2 (154.7)	0.267
GGT, Mean (SD)	125.9 (167.5)	467.9 (1239.5)	269.1 (816.7)	0.431
WBC count, Mean (SD)	10.3 (4.8)	12.6 (8.7)	11.3 (6.7)	0.659
Albumin level, Mean (SD)	33.8 (7.9)	29.8 (11.8)	32.1 (9.8)	0.438
Platelet count, Mean (SD)	298.2 (95.7)	265.5 (130.7)	284.9 (110.9)	0.159
INR, Mean (SD)	1.5 (2.5)	1.1 (0.11)	1.4 (1.9)	0.586

* Radiological details were missing for one patient in the reconstituting group. ** Gallbladder wall thickness was determined based on preoperative radiological imaging (ultrasound or CT). *** Normal laboratory reference ranges were as follows: total bilirubin (0–21 μmol/L), direct bilirubin (0–7 μmol/L), ALP (40–130 U/L), GGT (5–55 U/L), WBC (4–11 × 10^9^/L), albumin (35–50 g/L), INR (0.9–1.1), and platelet count (150–400 × 10^9^/L). ALP: Alkaline phosphatase; GGT: Gamma-glutamyl transferase; INR: International normalized ratio; SD: Standard deviation; WBC: White blood cell count.

**Table 4 healthcare-13-02465-t004:** Intra- and postoperative findings of the included patients.

	Fenestrating (*n* = 26)	Reconstituting (*n* = 20)	Total (*n* = 46)	*p*-Value
Operative Difficulty				0.784
Difficult	6 (23.1%)	4 (20%)	10 (21.7%)	
Very Difficult	20 (76.9%)	16 (80%)	36 (78.3%)	
Operative Duration (min), Mean (SD)	181.7 (64.2)	191.8 (73.5)	186.1 (67.7)	0.65
Blood Loss (mL), Mean (SD)	153.9 (104.1)	121.7 (105.7)	139.1 (104)	0.347
BDI	2 (7.7%)	0	2 (4.3%)	0.21
Intraoperative Drain Placement	23 (88.5%)	15 (75%)	38 (82.6%)	0.232
Peri-cholecystic Fluid Collection	6 (23.1%)	8 (40%)	14 (30.4%)	0.216
Presence of Bile Leak	6 (23.1%)	2 (10%)	8 (17.4%)	0.246
Previous History of Biliary Inflammation (Cholecystitis, Cholangitis)	7 (26.9%)	8 (40%)	15 (32.6%)	0.348
Hospital Stay (days), Mean (SD)	9 (3.3)	5 (4.6)	7.3 (4.4)	0.977
Follow-Up (months), Mean (SD)	58.3 (86)	31.2 (28.4)	46.5 (68.1)	0.144

BDI: Bile duct injury; SD: Standard deviation.

## Data Availability

Data are available from the corresponding author upon reasonable request.

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
