# Peer review of "Outcomes of Fenestrating vs. Reconstituting Laparoscopic Subtotal Cholecystectomy: A Single-Center Retrospective Study"

_healthcare, 2025, doi:10.3390/healthcare13192465_

Round 1
Reviewer 1 Report (Previous Reviewer 1)
Comments and Suggestions for Authors
Congratulations to the authors for the submitted manuscript, as it complies with the ethical guidelines and policies, is appropriate for the scope of the journal, the figures do not require clarification, all data are available in the tables, statistical tests are adequate, and the linguistic quality is suitable for peer review.
After the corrections made, I believe the manuscript meets the requirements for acceptance for publication.
Author Response
Thank you for taking the time to review our manuscript. Your input has been extremely valuable.
Reviewer 2 Report (Previous Reviewer 2)
Comments and Suggestions for Authors
The authors have made the corrections I mentioned.
Recommendations:
1. Even in line 233 of the Discussion section, the rate of bile leakage is 19.2%, while in Table 4, this rate is reported as 23.1%. This should be corrected.
Author Response
Thank you for your comment. We apologize for this oversight. We have performed a full audit of all reported counts, percentages and p-values and corrected inconsistencies and typographical errors.
Reviewer 3 Report (New Reviewer)
Comments and Suggestions for Authors
Dear Authors,
The title is not emphasized well; it should be clarified in the title that the study is retrospective and of single center. The introduction is well written. It is clearly written in the Methods that one of the exclusion criteria was conversion to open cholecystectomy. Why, then, in Table 1, in the description of the parameters of a very difficult operation, conversion is also described? Aren't these patients excluded from the study? The limitations are well explained.
Author Response
Thank you very much for your astute comment. We agree, our paper explicitly excludes patients that were converted to open. With that being said, the description parameter for table 1 was created by other well-established papers (cited in the references). In their analysis of “difficult operation” they included conversion to open as one of the variables. We merely adopted and cited their criteria for operative difficulty.
We have added this paragraph to the methods section to better explain our included patients: "Moreover, we excluded cases converted to open cholecystectomy as the objective of the present analysis was to describe outcomes of laparoscopic subtotal cholecystectomy performed and completed laparoscopically. We acknowledge that this exclusion may reduce the representation of the most severe cases and therefore introduces selection bias.
The two LSTC subtypes are described as ‘fenestrating’ (open remnant without restoration of lumen) and ‘reconstituting’ (closure of the remnant, also referred to here as ‘reconstituting [closure] LSTC’ or ‘closure after partial resection’).”
Reviewer 4 Report (New Reviewer)
Comments and Suggestions for Authors
Recommendations for the Authors
-
Reframe the study: Acknowledge the retrospective nature and the inherent limitations more prominently. The manuscript should be reframed as a descriptive study of a single institution's experience rather than a comparative study aiming to prove equivalence.
-
Correct data and formatting: Carefully recheck all patient counts, percentages, and p-values in the abstract, results section, and tables for consistency. Redraw the tables to be clearer and more professional. Ensure that all percentages are calculated correctly and add up to 100% where appropriate.
-
Revise the statistical analysis: Acknowledge that the study is underpowered and that a direct comparison is inappropriate without adjusting for baseline differences. Remove any language that suggests a "trend" or "slight increase" for non-significant findings. This study cannot conclude that the two techniques are comparable.
-
Strengthen the "Limitations" section: Expand on the confounding by indication. Explain that the decision to use one technique over another was not random and may have been influenced by the difficulty of the case. Clearly state that the small sample size precludes any definitive conclusions about the safety or efficacy of one technique over the other.
-
Update the text: Correct all typos and grammatical errors. Ensure that the use of terminology is consistent throughout the manuscript (e.g., "Forty-six" vs. "Fourty-six").
Author Response
Response to Reviewer 4
Reviewer comment 1 Reframe the study: Acknowledge the retrospective nature and the inherent limitations more prominently. The manuscript should be reframed as a descriptive study of a single institution's experience rather than a comparative study aiming to prove equivalence.
Authors’ response: Thank you very much for your comment. We completely agree. Based on your suggestion, we have reframed manuscript into a descriptive study rather than a comparison study. The title now emphasizes its retrospective, single-center, descriptive nature and we have removed wording implying a head-to-head proof of equivalence from the manuscript. Furthermore, in the text, we explicitly state that the study is descriptive and underpowered for formal comparison or equivalence testing (see Limitations and Conclusion sections).
Discussion/Limitation: “Additionally, this study is underpowered due to the limited sample size, which increases the risk of type II errors. Given the retrospective, single-center design and surgeon-dependent decision-making, confounding by indication is likely. Therefore, results should be better interpreted as descriptive rather than comparative.”
Conclusion: “Nevertheless, because of the retrospective design, small sample size, and lack of randomization or multivariable adjustment, no definitive conclusions regarding equivalence or superiority between the two techniques can be drawn.”
Reviewer comment 2 Correct data and formatting: Carefully recheck all patient counts, percentages, and p-values in the abstract, results section, and tables for consistency. Redraw the tables to be clearer and more professional. Ensure that all percentages are calculated correctly and add up to 100% where appropriate.
Authors’ response: Thank you, we performed a full audit of all reported counts, percentages and p-values and corrected inconsistencies and typographical errors.
- Tokyo Grade 2 in the fenestrating group: 15 (57.7%) (corrected from earlier inconsistent figure 56.5% in Table 2);
- Discussion paragraph previously quoted p = 0.388 for bile leak as was a typographical error; the correct p-value is p = 0.246 (as correctly reported in Table 4). We corrected the Discussion to use p = 0.246.
- Table 4: “Very difficult” for fenestrating: corrected percent 20/26 = 76.9% (was incorrectly printed as 66.9%).
- Combined (total) hospital stay column: original manuscript reported 16.8 (35.9) which is incorrect. Recomputed pooled value is mean = 7.3 days (SD 4). Table 4 updated accordingly.
- Discussion paragraph previously incorrectly quoted bile leakage as 19.2%; correct value was 23.1%.
Additionally, minor typographical and formatting errors in the Results and Discussion were corrected (see corrected examples above).
Reviewer comment 3 Revise the statistical analysis: Acknowledge that the study is underpowered and that a direct comparison is inappropriate without adjusting for baseline differences. Remove any language that suggests a "trend" or "slight increase" for non-significant findings. This study cannot conclude that the two techniques are comparable.
Authors’ response: We accepted this excellent recommendation. All phrases implying a “trend”, “slight increase” or suggesting comparability based on non-significant differences have been removed or rephrased to be conservative.
Manuscript changes: In the Abstract:
“Fenestrating procedures had had a numerically higher incidence, though not statistically significant, incidence of BDI (7.7% vs. 0%; p = 0.21), bile leakage (23.1% vs. 10%; p = 0.246), and intraoperative drain placement (88.5% vs. 75%; p = 0.232). Postoperative complications, such as bile leaks, were also comparable between the two techniques. Nevertheless, given the small sample, these observations are descriptive and should not be interpreted as evidence of comparability or superiority.”
Furthermore, we added a sentence in Methods/Statistics clarifying that no multivariable regression was performed because of limited sample size, and therefore we cannot exclude residual confounding.
Reviewer comment 4 Strengthen the "Limitations" section: Expand on the confounding by indication. Explain that the decision to use one technique over another was not random and may have been influenced by the difficulty of the case. Clearly state that the small sample size precludes any definitive conclusions about the safety or efficacy of one technique over the other.
Authors’ response: Thank you for this astute comment. We substantially expanded the Limitations section to explicitly discuss confounding by indication, surgeon decision bias, the exclusion of conversions (selection bias), variability in intraoperative technique and follow-up duration, and the underpowered nature of the study. We underlined that:
“For these reasons, our results should be interpreted as descriptive and hypothesis-generating rather than definitive evidence of equivalence or superiority between techniques.”
Reviewer comment 5. Update text: correct typos and consistent terminology.
Authors’ response: All typographical errors and inconsistent wordings were corrected (examples: fixed any occurrences of “Fourty-six” → “Forty-six”; standardized use of numerals for counts and percentages; fixed missing decimal points and p-value formatting such as p =0.027 → p = 0.027). Terminology (fenestrating / reconstituting) is now used consistently (see below we document a small terminology change suggested by Reviewer 5 as well).
Reviewer 5 Report (New Reviewer)
Comments and Suggestions for Authors
The article we have to review is entitled "outcomes of fenestrating vs recontituting laparoscopic subtotal cholecystectomy".We have received a version with corrections in the body of the text. This text may have already been reviewed by another reader.
Regarding the background
Several biases related to the retrospective nature of the study should be highlighted: the exclusion of cholecystectomies converted to laparotomy; the difference in severity of damage between fenestrated forms, which are more severe than forms with reconstruction.
The ‘very difficult’ group includes conversions that were subsequently excluded from the study. It represents a major bias and should be explained.
The conclusion in the abstract is not related to the study comparing the two techniques. The conclusion that bailout LSTCs are safe is true but not illustrated by this study; only the following part reflects the retrospective study: our analysis demonstrates that fenestrating...
Regarding the form
The term ‘reconstituting laparoscopic subtotal cholecystectomy’ is rarely used in the literature. While it can be used in the body of the text if defined beforehand, it would be clearer to use another term in the text, such as ‘fenestrating vs closure after partial resection’.
The phrase ‘our study demonstrates that fenestrating...’ used in the conclusion is strong terminology for a retrospective study. The use of ‘tends to demonstrate’ instead of ‘demonstrates’ would be more appropriate.
In the introduction, lines 59 to 65, the disadvantages of the two techniques should be presented more clearly: for fenestration, the theoretical disadvantage is that it can lead to bile leakage; in closure and therefore reconstruction of the gallbladder, the disadvantage is that the gallbladder is pathological at the end of the procedure, which may lead to the formation of new gallstones and all their complications. The recurrence is a major risk in this group and this recurrence has appeared in the study (line 248) and with a longest follow up it will be probably appearing more frequently. This point of the follow up and the wide variation of follow up is clearly explain by the authors line 287 which is a good point.
The conclusion in the text is clearly more convenient than the one in the abstract.
Author Response
Response to Reviewer 5
Reviewer comment The article we have to review is entitled "outcomes of fenestrating vs recontituting laparoscopic subtotal cholecystectomy".We have received a version with corrections in the body of the text. This text may have already been reviewed by another reader.
Regarding the background Several biases related to the retrospective nature of the study should be highlighted: the exclusion of cholecystectomies converted to laparotomy; the difference in severity of damage between fenestrated forms, which are more severe than forms with reconstruction.
Authors’ response: We thank the reviewer for highlighting these important biases. We added explicit text in Methods and Limitations noting that converted cases were excluded (rationale: to evaluate laparoscopic subtotal technique outcomes specifically) and that this exclusion likely biased the sample toward cases in which LSTC was completed laparoscopically.
In Methods we stated that:
“Patients were excluded if they had incomplete medical records or had malignant gallbladder disease. Moreover, we excluded cases converted to open cholecystectomy as the objective of the present analysis was to describe outcomes of laparoscopic subtotal cholecystectomy performed and completed laparoscopically. We acknowledge that this exclusion may reduce the representation of the most severe cases and therefore introduces selection bias.”
The Limitations and Conclusion sections were completely revised (see response to reviewer 4 and the improved manuscript).
Reviewer comment The ‘very difficult’ group includes conversions that were subsequently excluded from the study. It represents a major bias and should be explained. The conclusion in the abstract is not related to the study comparing the two techniques. The conclusion that bailout LSTCs are safe is true but not illustrated by this study; only the following part reflects the retrospective study: our analysis demonstrates that fenestrating...
Authors’ response: Thank you for your astute comment. We clarified the workflow in Methods: the preoperative and intraoperative difficulty scores were applied to all attempted laparoscopic cases at the time of the operation, but only patients who ultimately underwent laparoscopic subtotal cholecystectomy and were not converted were included in the analytic dataset. We added a sentence explaining that therefore the distribution of “very difficult” cases in the analyzed sample may underrepresent the true severity distribution among all attempted cases (since some conversions—often the most severe—were excluded). This is now explicitly described in Methods and Limitations.
“Operative difficulty scores were recorded prospectively in the operative note for all attempted laparoscopic cholecystectomies; however, this analysis includes only patients in whom LSTC was completed laparoscopically. Consequently, the proportions of ‘very difficult’ cases reported here exclude converted cases and may not reflect the full spectrum of difficulty encountered during attempted laparoscopic cholecystectomy.”
Reviewer comment The conclusion in the abstract is not related to the study comparing the two techniques. The conclusion that bailout LSTCs are safe is true but not illustrated by this study; only the following part reflects the retrospective study: our analysis demonstrates that fenestrating..
Authors’ response: Thank you for raising this excellent point. We revised the Abstract conclusion to remove any strong statements about safety or comparability and to align the Abstract with the descriptive nature of the data. (See the Abstract wording proposed above under Reviewer 4.)
“Conclusions: Despite limitations, our analysis suggests that fenestrating and reconstituting approaches have comparable postoperative outcomes, though fenestrating procedures were associated with a slight increase in BDI and drain placement due to leaks. The choice of the LSTC technique should depend on intraoperative findings and surgical expertise, and familiarity with each technique, but further studies are needed to obtain firm conclusions.”
Reviewer comment The term ‘reconstituting laparoscopic subtotal cholecystectomy’ is rarely used in the literature. While it can be used in the body of the text if defined beforehand, it would be clearer to use another term in the text, such as ‘fenestrating vs closure after partial resection’.
Authors’ response: We agree. At first mention we now define the term and introduce a clear short label to be used consistently: “reconstituting (closure) LSTC” and we optionally provide the synonym “closure after partial resection” in parentheses. Thereafter we use the short forms: “fenestrating LSTC” and “reconstituting (closure) LSTC”. In the Methods:
“The two LSTC subtypes are described as ‘fenestrating’ (open remnant without restoration of lumen) and ‘reconstituting’ (closure of the remnant, also referred to here as ‘reconstituting [closure] LSTC’ or ‘closure after partial resection’).”
Reviewer comment The phrase ‘our study demonstrates that fenestrating...’ used in the conclusion is strong terminology for a retrospective study. The use of ‘tends to demonstrate’ instead of ‘demonstrates’ would be more appropriate.
Authors’ response: We replaced strong causal/demonstrative verbs throughout (e.g., “demonstrates”) with conservative alternatives such as “suggests,”, and as indicated by another reviewer. Therefore, we explicitly state that findings are descriptive and hypothesis generating.
Reviewer comment In the introduction, lines 59 to 65, the disadvantages of the two techniques should be presented more clearly: for fenestration, the theoretical disadvantage is that it can lead to bile leakage; in closure and therefore reconstruction of the gallbladder, the disadvantage is that the gallbladder is pathological at the end of the procedure, which may lead to the formation of new gallstones and all their complications.
Authors’ response: Thank you for this valuable addition. We rewrote the relevant sentences to clearly state the theoretical and observed disadvantages: fenestration → bile leak; reconstitution → pathological remnant with risk of recurrent stones/symptoms. Introduction:
“Fenestrating LSTC avoids creation of a closed remnant but may increase the risk of postoperative bile leakage from the open gallbladder base. In contrast, reconstituting (closure) LSTC seals the gallbladder remnant and may reduce immediate bile leakage, but it leaves a pathological remnant that can act as a nidus for recurrent stone formation and recurrent biliary symptoms over time. The long-term risk of symptomatic remnant disease is therefore a key concern with the reconstituting approach.”
Reviewer comment The recurrence is a major risk in this group and this recurrence has appeared in the study (line 248) and with a longest follow up it will be probably appearing more frequently. This point of the follow up and the wide variation of follow up is clearly explain by the authors line 287 which is a good point.
Authors’ response: We agree and expanded Limitations to emphasize wide variability in follow-up duration (SD large), lack of systematic radiological or endoscopic surveillance, and consequent risk of under-ascertaining late remnant complications. We recommend prospective multicenter registries with standardized imaging/follow-up to address this.
Round 2
Reviewer 4 Report (New Reviewer)
Comments and Suggestions for Authors
The authors of the manuscript have made a perfect effort to address all major and minor concerns raised during the initial review. The revisions are thorough, well-reasoned, and have significantly improved the quality and clarity of the manuscript.
This manuscript is a resubmission of an earlier submission. The following is a list of the peer review reports and author responses from that submission.
Round 1
Reviewer 1 Report
Comments and Suggestions for Authors
Congratulations to the authors for the submitted manuscript, as it complies with the ethical guidelines and policies, is appropriate for the journal's scope, the figures do not require clarification, all data are available in the tables, the statistical tests are adequate, and the linguistic quality is adequate for peer review.
I have a number of questions that I don't know if they can be clarified. How many surgeons performed the surgeries? Do they all have similar experience performing cholecystectomies? In the case of elective surgery, how long did it take from the acute episode to the actual procedure? Does the number of previous cholecystectomy episodes influence the need for one or the other of these techniques?
Table 2 specifies the type of cholecystitis (acute, acute on chronic, and chronic) and the type of surgery (elective and emergent). Could the type of cholecystitis be better correlated with the type of surgery? As described in the literature, the time elapsed between the onset of symptoms and the performance of surgery can increase the difficulty of the procedure.
Regarding the bibliographic references, although they are adequate, I believe they could be improved with additional systematic reviews and meta-analyses, comparative studies, etc.
There are limitations, as the authors note. The sample size is small, the study is retrospective, and the data obtained can sometimes be biased.
Reviewer 2 Report
Comments and Suggestions for Authors
The authors retrospectively compared fenestration and reconstruction methods in 46 difficult laparoscopic subtotal cholecystectomy cases.
Recommendations;
1. Did the same team perform these surgeries? It should be stated in the Method section.
2. Was the cystic duct sutured from the inside in all patients who underwent fenestration? It should be stated.
3. I think the number of reconstitution emergencies in Table 2 will be 9. Otherwise, the total number of patients will be 30. It should be corrected.
4. Tokyo Guidelines Grading should be checked in the fenestration section in Table 2. According to the table, the total number of patients is 21 and the total number of patients is 43. It should be corrected.
5. Clinical should be added to the title of Table 3. Because the clinical palpable gallbladder parameter is used in the table.
6. How was the gallbladder wall thickness determined in Table 3 and the text? It should be explained.
7. The working ranges of the biochemical parameters reported in Table 3 should be specified.
8. The biochemical parameters reported in Table 3 cannot be total results, but they can be mean values. They should be corrected.
9. Are there any patients who underwent preoperative ERCP or PTC in the patient group with such high bilirubin values? They should be specified.
10. Bile leakage was mentioned twice in the discussion section, and it happened again. It should be corrected.
Reviewer 3 Report
Comments and Suggestions for Authors
- This study employed a single-center retrospective cohort design, which is inherently subject to information bias and selection bias. The reliance on existing medical records may lead to inconsistencies in data recording and missing variables. Furthermore, retrospective analyses cannot establish causal relationships, only associations.
- Only 46 patients were included in total (20 in the reconstituting group and 26 in the fenestrating group). Such a small sample size limits the statistical power to detect significant differences, particularly for rare complications such as bile duct injury (BDI) or bile leakage. The underpowered sample may lead to type II errors.
- All patients were recruited from a single tertiary care center in Saudi Arabia. Differences in surgical expertise, patient demographics, and healthcare infrastructure may limit the external validity and generalizability of the findings to other institutions or countries.
- The choice between fenestrating and reconstituting techniques was determined by intraoperative judgment and surgeon preference, without randomization or standardized selection criteria. This introduces the risk of confounding by indication, where sicker or more complex cases may have been selectively assigned to one technique over the other.
- Although the mean follow-up period was 46.5 months, the standard deviation was large (68.1 months), indicating substantial variation among patients. This heterogeneity may lead to underreporting of long-term complications such as remnant gallbladder inflammation, stone formation, or recurrent symptoms.
- The study focused solely on clinical and surgical metrics such as operative duration and postoperative complications. It did not evaluate patient-centered outcomes like postoperative pain, quality of life (QoL), or return to normal function, which are essential for a comprehensive assessment of surgical impact.
- Although the two techniques were defined, variations in intraoperative details—such as closure of the gallbladder remnant, cystic duct management, or drain placement—were not standardized and likely varied among surgeons. These inconsistencies may have influenced outcomes and reduce internal validity.
- Only univariate comparisons were performed. Important covariates such as BMI, ASA score, gallbladder wall thickness, and preoperative severity grading were not adjusted for using multivariable models, potentially confounding the observed associations.
- Although a few cases of symptom recurrence were mentioned, the study did not report whether patients required postoperative interventions such as ERCP, percutaneous drainage, or reoperation. These data are critical for evaluating surgical durability and downstream resource utilization.
- The reconstituting technique retains a portion of the gallbladder, which may carry long-term risks such as inflammation, stone formation, or even neoplastic transformation. The study did not include long-term imaging follow-up or surveillance for remnant-related complications, leaving this issue unaddressed.
The English could be improved to more clearly express the research.
